# Molecular Characterization of a DNA Polymerase from *Thermus thermophilus* MAT72 Phage vB_Tt72: A Novel Type-A Family Enzyme with Strong Proofreading Activity

**DOI:** 10.3390/ijms23147945

**Published:** 2022-07-19

**Authors:** Sebastian Dorawa, Olesia Werbowy, Magdalena Plotka, Anna-Karina Kaczorowska, Joanna Makowska, Lukasz P. Kozlowski, Olafur H. Fridjonsson, Gudmundur O. Hreggvidsson, Arnthór Aevarsson, Tadeusz Kaczorowski

**Affiliations:** 1Laboratory of Extremophiles Biology, Department of Microbiology, Faculty of Biology, University of Gdansk, 80-308 Gdansk, Poland; sebastian.dorawa@ug.edu.pl (S.D.); olesia.werbowy@ug.edu.pl (O.W.); magdalena.plotka@ug.edu.pl (M.P.); 2Collection of Plasmids and Microorganisms, Faculty of Biology, University of Gdansk, 80-308 Gdansk, Poland; anna.kaczorowska@ug.edu.pl; 3Department of General and Inorganic Chemistry, Faculty of Chemistry, University of Gdansk, 80-308 Gdansk, Poland; joanna.makowska@ug.edu.pl; 4Institute of Informatics, Faculty of Mathematics, Informatics and Mechanics, University of Warsaw, 02-097 Warsaw, Poland; lukasz.kozlowski.lpk@gmail.com; 5Matis, 113 Reykjavik, Iceland; olafur@matis.is (O.H.F.); gudmundo@matis.is (G.O.H.); arnthor@matis.is (A.A.); 6Department of Biology, School of Engineering and Natural Sciences, University of Iceland, 102 Reykjavik, Iceland

**Keywords:** DNA polymerase, *Thermus* phage, 3′-5′ exonuclease activity

## Abstract

We present a structural and functional analysis of the DNA polymerase of thermophilic *Thermus thermophilus* MAT72 phage vB_Tt72. The enzyme shows low sequence identity (<30%) to the members of the type-A family of DNA polymerases, except for two yet uncharacterized DNA polymerases of *T. thermophilus* phages: φYS40 (91%) and φTMA (90%). The Tt72 *polA* gene does not complement the *Escherichia coli*
*polA*^−^ mutant in replicating *polA*-dependent plasmid replicons. It encodes a 703-aa protein with a predicted molecular weight of 80,490 and an isoelectric point of 5.49. The enzyme contains a nucleotidyltransferase domain and a 3′-5′ exonuclease domain that is engaged in proofreading. Recombinant enzyme with His-tag at the *N*-terminus was overproduced in *E. coli*, subsequently purified by immobilized metal affinity chromatography, and biochemically characterized. The enzyme exists in solution in monomeric form and shows optimum activity at pH 8.5, 25 mM KCl, and 0.5 mM Mg^2+^. Site-directed analysis proved that highly-conserved residues D15, E17, D78, D180, and D184 in 3′-5′ exonuclease and D384 and D615 in the nucleotidyltransferase domain are critical for the enzyme’s activity. Despite the source of origin, the Tt72 DNA polymerase has not proven to be highly thermoresistant, with a temperature optimum at 55 °C. Above 60 °C, the rapid loss of function follows with no activity > 75 °C. However, during heat treatment (10 min at 75 °C), trehalose, trimethylamine *N*-oxide, and betaine protected the enzyme against thermal inactivation. A midpoint of thermal denaturation at T_m_ = 74.6 °C (ΔH_cal_ = 2.05 × 10^4^ cal mol^−1^) and circular dichroism spectra > 60 °C indicate the enzyme’s moderate thermal stability.

## 1. Introduction

DNA replication, an essential feature of all life forms, relies on a set of specialized proteins that, complexed in the replisome network, can coordinate and execute the entire multi-step process in a highly concerted and accurate manner [1,2,3,4]. A central role in this molecular machinery is played by DNA polymerases, which are very diverse multifunctional enzymatic entities that copy, maintain, and faithfully transmit genetic information through generations [5,6,7]. They are also involved in repair and recombination, which are crucial mechanisms that control genomic DNA stability and drive genetic variation [8,9,10]. DNA polymerases’ principal function relies on the efficient and faithful template-directed incorporation of complementary 2′-deoxyribonucleotides to synthesized primer strands in a a 5′-3′ direction. Based on their molecular structure, functional features, and amino acid sequence similarity, DNA polymerases fall into seven families: A, B, C, D, X, Y, and RT [11,12,13,14,15]. Despite observed striking differences, especially at the amino acid sequence level [16], there is a functional equivalency between DNA polymerases suggested by in vivo complementation between enzymes from organisms that are phylogenetically distant [17,18].

The type-A family consists of enzymes (Pol I, or PolA-type) spread widely among various microbes, including thermophiles and bacteriophages [19,20,21]. A recent analysis revealed that approximately 25% of known double-stranded DNA phages encode the *polA* gene [22]. A prototype enzyme of this family is DNA polymerase I of *Escherichia coli* (Eco pol I). It consists of a single polypeptide chain with 928 amino acids possessing three distinct enzymatic activities: a 5′-3′ DNA nucleotidyltransferase, a 5′-3′ exonuclease (5′-3′ exo), and a 3′-5′ exonuclease (3′-5′ exo, proofreading). Each of them is associated with a separate structural subunit. Such a modular structure implies the precise coordination of all enzyme functions [23]. A critical role is played by a 3′-5′ exo domain involved in the proofreading mechanism, a critical checkpoint ensuring a low error rate in a newly-synthesized DNA strand (10^−8^–10^−10^) [24]. DNA polymerases belonging to A-family can also show additional features, such as reverse transcriptase activity (*Tth* DNA polymerase; [25]), strand displacement activity (*Thermus brockianus* DNA polymerase; [26]), or the ability to efficiently incorporate 2′,3′-dideoxynucleotides (phage T7 DNA polymerase; [27]). 

Enzymes representing A-family play a minor role in replicating bacterial chromosomes (lagging-strand synthesis and short patch repair); however, their function is crucial in replicating the ColE1-type plasmids of enterobacteria [28,29]. On the other hand, type-A phage DNA polymerases, exemplified by the phage T7 enzyme, are heavily engaged in viral DNA synthesis in a highly processive manner [30,31].

The multifunctionality of DNA polymerases has made them indispensable tools in molecular biology [32]. Notably, they found many applications in molecular diagnostics and enabled the development of unique technologies for life sciences. For example, the heat-stable *Taq* DNA polymerase (Taq pol I) of a thermophilic bacterium *Thermus aquaticus* has become a workhorse of PCR-based DNA amplification [33]. Another enzyme, the large fragment of *Bst* DNA polymerase from *Geobacillus stearothermophilus,* found application in the isothermal amplification of specific DNA fragments [34,35]. However, there is still a need for novel enzymes with high innovation value for bioscience and diagnostic applications. This quest is particularly urgent in SARS-CoV-2 pandemics, when the development of new genetic tools is of utmost importance. Consequently, many efforts focus on searching for new DNA polymerases, especially from microbes or metagenomes that have been isolated from extreme habitats [36,37]. 

Our work focuses on viruses and viral metagenomes from extreme hydrothermal habitats [38]. Geochemical hot spots are rich in viruses [39,40]. However, their genomes bear little or no similarity to known nucleotide sequences deposited in the GenBank non-redundant database, making them hard to identify [41]. Thus, these relatively underexplored habitats offer immense genetic resources with prospects for new products and applications. Several novel *Thermus* phage proteins have been characterized in our laboratory within the VIRUS-X project [38]. They participate in DNA metabolism as Tt72RecA [42] and novel archaeal-like Holliday junction-resolving enzyme [43], or through cell lysis as endolysins [44,45,46,47]. Furthermore, the enzymes from phages are frequently more robust, simpler in structure, and more efficient than those from the native host. Although A-family DNA polymerases have been studied for decades and established some excellent research models to study nucleic acid metabolism (e.g., Eco pol I 5′-3′ exo^−^ (Klenow Fragment), phage T7 DNA polymerase), enzymes from thermophilic phages are poorly explored. The presented study has characterized an A-family DNA polymerase of phage vB_Tt72 (Tt72 pol) of the thermophilic eubacterium *Thermus thermophilus*. Like other viral DNA-dependent replicases, it possesses only a 3′-5′ exonuclease activity. Our analysis suggests that the vB_Tt72 phage, representing the *Myoviridae* family, is related to φYS40 [48] and φTMA [49], which infect the host bacterium, *T. thermophilus*. To our knowledge, this is the first report that characterizes a type-A family DNA polymerase from a thermophilic phage. 

## 2. Results

### 2.1. In Silico Analysis of T. thermophilus Phage vB_Tt72 Genome in Search for DNA Polymerase

A computational analysis of the genomic sequence of *T. thermophilus* phage vB_Tt72 (~154 kb) belonging to the *Myoviridae* family using the BlastP suite revealed an open reading frame that showed similarity to the type-A DNA polymerases. The corresponding nucleotide sequence has been deposited in GenBank (accession number ON714139). The Tt72 *polA* gene comprises 2112 nucleotides and codes for a 703-aa protein with a predicted molecular weight of 80,490 and an isoelectric point of 5.49. Upstream from the coding sequence there is a promoter with -10 and -35 sequences (-10, TATTAT; -35, TTGACA) that are characteristic of early promoters of *T. thermophilus* phage φYS40, which is homologous to phage Tt72 [50]. The gene starts with the initiation codon ATG and ends at the TAA termination codon. This gene’s overall guanine-plus-cytosine (G + C) content is 31.5%, which is much lower than the G + C content of *T. thermophilus* genomic DNA (69.5%; [51]). The codon usage of the Tt72 *polA* gene reflects its low G + C content, with 79.5% of the codons ending with either A or T. To determine the differences in the codon usage pattern of the Tt72 *polA* gene and that of *T. thermophilus*, we calculated the value of the codon adaptation index (CAI) using the JCat computer program [52]. Based on the assumption that genes with a high CAI (near 1) belong to the class of highly-expressed genes, we infer from the CAI value for the Tt72 *polA* gene, which is 0.04, its low rate of expression in *T. thermophilus*. 

As shown in Figure 1, the amino acid sequence of Tt72 pol at one point exhibits low similarity with other A-family members, such as DNA polymerases either from mesophilic or thermophilic sources: *E. coli* Pol I (GenBank WP_000250029.1; 31.1% identity, 66% query coverage), *E. coli* phage T5 (YP_006950.1; 26.8% identity, 42% query coverage), *E. coli* phage T7 (NP_041982.1; 24.1% identity, 22% query coverage), *T. thermophilus* HB8 (BAA06033.1; 31.5% identity, 51% query coverage), and *T. aquaticus* (AAA27507.1; 30.2% identity, 51% query coverage) (Figure 1a). On the other hand, the Tt72 pol amino acid sequence shares a high resemblance to the putative DNA polymerase derived from the Gongxiaoshe hot spring metagenome, China, Western Yunnan Province, Ruidian geothermal area (HHS22965.1; 60.3% identity, 98% query coverage; [53]) and two yet uncharacterized DNA polymerases of *T. thermophilus* phages: φYS40 (YP_874046.1; 91.3% identity, 100% query coverage) and φTMA (YP_004782240.1; 90.6% identity, 100% query coverage) were isolated from Mine and Otagawa hot springs, Japan [48,49,54]. A Pfam database search revealed that the Tt72 pol has a modular structure (Figure 1b) with two functional domains containing highly-conserved motifs that are characteristic of type-A family DNA polymerases [11,13]. These include three motifs (Exo I, Exo II, and Exo III; [55]) within a 201 amino acid stretch of the *N*-terminal domain with 3′-5′ exo activity and six motifs (1, 2a, 2b, 3, 4, 5, and 6; [14,56]) within the larger 502 amino acid *C*-terminal domain that has a characteristic nucleotidyltransferase fold.

### 2.2. Biological Activity of Tt72 DNA Polymerase in E. coli

The functionality of Tt72 pol was tested in an *E. coli polA*^−^ background. We used tthe *E. coli* JS200 *recA718 polA12*^ts^ strain encoding a temperature-sensitive Eco pol I. In rich media, this double-mutant cannot grow at 42 °C [57]. We also exploited the feature that the replication of the ColE1-type plasmids depends on the Eco pol I activity (Figure 2e). Therefore, the pBR322, a ColE1-type plasmid with a *polA*-dependent pMB1 origin of replication, was used in the presented experiment. To test the ability of Tt72 pol to complement the Eco pol I function, plasmid pHSG-polTt72 with a Tt72 *polA* gene under the control of the *tac* promoter was introduced into *E. coli* JS200 [pBR322] and transformants were grown at 30 °C and 42 °C. The test revealed that the Tt72 pol could not complement the *E. coli polA* gene (Figure 2b). In control experiments, we observed complementation when plasmids with cloned DNA polymerases were used, such as pHSG_pol_I (Eco pol I), pHSG_polTth (Tth pol), or pHSG_polTaq (Taq pol I) (Figure 2a,c,d).

### 2.3. Overproduction and Purification of Tt72 DNA Polymerase

The Tt72 *polA* gene was cloned into the pET15b expression vector under the control of the T7 promoter to generate the pET15b_polTt72 plasmid. Tt72 pol was overproduced at 30 °C in *E. coli* BL21(DE3)[pRARE] harboring pET15b_polTt72 (Figure 3a and Appendix A). The pRARE plasmid (Novagen, Madison, WI, USA) was used to improve protein overproduction as it carries genes for the host’s rare codon tRNAs (Arg, Gly, Ile, and Pro) and thus optimizes codon usage in heterologous genes [58]. However, the yield of a recombinant protein produced in a soluble form was low (Appendix A). The problem could be overcome by lowering the culture temperature to 18 °C. In this setting, the overproduction gave a more soluble recombinant protein (Appendix A). The consecutive steps concerning the overproduction of Tt72 pol are shown in Figure 3. After induction of Tt72 pol production with IPTG, we observed bands corresponding to the recombinant enzyme and products of Tt72 pol degradation (Figure 3b, panels I and II). 

A three-step procedure was used to purify the recombinant Tt72 pol with His-tag at the *N*-terminus. It consisted of heat treatment, HiTrap^TM^ Talon column chromatography, and HiTrap^TM^ Heparin HP column affinity chromatography. At each step, the purity of the enzyme preparation was monitored by SDS-PAGE (Appendix A). The homogeneity and molecular weight of recombinant protein were determined by size exclusion chromatography on a Superdex 75 10/300 GL column (GE Healthcare, Uppsala, Sweden). The Tt72 pol was eluted in a single peak at a volume of 9.46 mL, which corresponds to a molecular mass of 83.7 kDa (Figure 4a). The determined molecular mass is slightly higher than predicted from the amino acid sequence of the recombinant enzyme (82.7 kDa). As shown in Figure 4b, an SDS-PAGE analysis of the purified Tt72 pol preparation showed additional ~45 kDa and ~37 kDa bands. To determine whether these bands are *bona fide* products of degradation of Tt72 pol, western blot analysis with anti-His-tag antibodies was performed. This analysis revealed that the additional band (~37 kDa) contains an *N*-terminal portion of Tt72 pol (Figure 4b). In the next step, the gene coding for Tt72 pol was cloned into pET24a(+) to produce a protein with *C*-terminal His-tag. Then, after purification, the western blot analysis was repeated to reveal traces of degradation products (Figure 4c). We strengthen the cell lysis conditions by increasing the salt concentration to 500 mM NaCl to prevent enzyme degradation. We found this step effective based on our experience with another enzyme, FokI DNA methyltransferase, which is prone to degradation during the standard purification procedure [59]. Consequently, we could minimize contamination of the final Tt72 pol preparation with degradation products (Appendix A, lane 6). From 4 g of bacteria (1 L of culture), we obtained 4.0 mg of Tt72 pol. In the next step, the purity of the enzyme and its oligomeric state was determined by analytical ultracentrifugation. It allowed us to conclude that Tt72 pol preparation is homogenous, and the enzyme exists in solution in a monomeric form (Appendix A).

### 2.4. 3′-5′ Exonuclease Activity of the Tt72 DNA Polymerase

The Tt72 pol contains the highly-conserved motifs (Exo I, Exo II, and Exo III), characteristic of enzymes with 3′-5′ exonuclease activity. The activity was assayed with the use of the 3′end-labeled DNA restriction fragments. In the case of Tt72 pol, we observed strong exonucleolytic activity, which was indicated by the release of about 95% of the [^3^H] from the 3′-end of the substrate DNA within 30 min in the absence of dNTPs and about 44% of the [^3^H] in the presence of dNTPs at a concentration of 100 µM in the reaction mixture (Figure 5). This data indicates that Tt72 pol possesses a strong 3′-5′ exo activity. The same strong 3′-5′ exo activity in the presence of dNTP was observed in the case of the DNA polymerase of the *T. thermophilus* φYS40 phage that is homologous to Tt72 pol [48].

### 2.5. Tt72 DNA Polymerase Lacks Terminal Transferase Activity

Some DNA polymerases exemplified by the enzymes from *T. aquaticus* and *T. thermophilus* have non-template-dependent activity that catalyze the addition of single adenine nucleotides at the 3′ end of a double-stranded DNA molecule [60]. Here, we tested the ability of the Tt72 pol to add an extra nucleotide at the 3’ end of DNA. For this purpose, we used a blunt-ended double-stranded 24-bp oligonucleotide with one strand labelled with Cy3 fluorophore at the 5′ end (Figure 6a). Such a double-stranded oligo was incubated with Tt72 pol in a reaction mixture supplemented with dNTPs. After completing the reaction, the extension products were analyzed on a 15% polyacrylamide/8 M urea gel. The result showed that the Tt72 pol lacks terminal transferase activity (Figure 6b), which is similar to Pfu DNA polymerase (Figure 6c). The control experiment showed that Taq pol I can add one extra nucleotide into a synthesized strand (Figure 6d). 

### 2.6. Validation of Catalytic Sites for Tt72 pol 3′-5′ Exo and Nucleotidyltransferase Domain

A detailed analysis of the homology-based model of the Tt72 pol revealed the putative spatial architecture of an enzyme with two functional domains (Figure 7a). The smaller, globular *N*-terminal domain (aa 1-201, Appendix A) comprises three β-strands and six α-helices. It contains a catalytic site for 3′-5′ exo composed, as in other type-A family enzymes, of four well-defined conserved acidic residues that are part of motifs Exo I, Exo II, and Exo III (D15, E17, D78, and D184; Figure 7). For 3′-5′ exo function, two other residues also seem to be critical: Y180 and L27; they correspond to Y497 and L361 of canonical *E. coli* DNA pol I. By its hydroxyl group, the former interacts with phosphate oxygen of a misincorporated nucleotide, while the latter stabilizes the site for nucleotide binding by hydrophobic interactions with the side chain [55]. The conservancy of residues involved in the 3’-5’ exonuclease active site suggests that the activity of Tt72 DNA pol relies on a two-metal-ion mechanism reaction to cleave the phosphodiester backbone that seems to be standard for phosphoryl-transfer enzymes [61]. By analogy to other type-A enzymes, it was proposed that this catalytic site is built of D15, E17, D78, Y180, and D184 (corresponding to D355, E357, D424, Y497, and D501 in Eco pol I). The importance of conserved residues was tested with the use of substitution variants. We found that substitutions within conserved residues are critical for 3′-5′ exonucleolytic activity and resulted in variants with impaired 3′-5′ exo activity compared to the wild-type enzyme: D15A (8 ± 0.1%), E17A (7 ± 3%), L27A (23 ± 0.1%), D78A (4 ± 1%), Y180A (4 ± 0.4%), and D184A (3 ± 1%) (Figure 7b). On the other hand, the nucleotidyltransferase activity of these variants was still high, exceeding 85% of the wild-type enzyme (Figure 7a). The binding capacity of these variants is also strongly affected. The dissociation constant (K_D_) measurements were done using microscale thermophoresis (Figure 7b). While the K_D_ for the wild-type enzyme was determined to be 2 ± 0.1 nM, the K_D_ values for substitution variants are much higher, indicating a low affinity for substrate oligo: D15A (232 ± 35 nM), E17A (529 ± 43 nM), L27A (32 ± 1 nM), D78A (109 ± 5 nM), Y180A (229 ± 33 nM), and D184A (144 ± 9 nM). 

The larger nucleotidyltransferase domain (aa 202-703, Appendix A) comprises six β-strands and twenty-one α-helices. It adopts the shape of a right hand, which is universal among other DNA polymerases. This anatomical analogy leads to three subdomains: the palm, thumb, and fingers. Their spatial arrangement contributes to the prominent cleft that houses the polymerase active site. In this respect, a unique role is played by the palm subdomain that is made of a six-stranded sheet that forms the base of the cleft. The most crucial active site residues in the model enzyme, Eco pol I, are D705, E710, D882, and E883, with carboxylate side chains. Across DNA polymerase families, the first two residues are strictly conserved, and a third variable is an acidic residue (Asp or Glu). These residues are part of conserved motifs 3 and 5 (palm) and coordinate two divalent metal ions involved in catalysis [30,62]. In the case of Tt72 pol, the respective catalytic residues seem to be D384, D615, and S616 (Figure 7a and Appendix A). The last one lacks the carboxylate side chain. We also tested E389 (E710 in Eco pol I)-neighboring D384, as shown later, to evaluate its importance in catalysis. Another essential element defines a DNA-binding groove, the fingers subdomain, which is structured by α-helices and forms a boundary on one cleft side. We noted the presence of an additional α–helix (α23, aa 526-554) missing in the ternary structure of Eco pol I and T7 pol (Figure 1a and Appendix A). The thumb subdomain closes the cleft with two long anti-parallel α-helices and two short α-helices at the tip (Figure 7a). They contain motifs 1, 2a, and 2b, which interact with the minor groove of the product, the double-stranded DNA that is upstream of the polymerase active site [63]. A similar role is associated with residues of motif 6 in the palm subdomain [56]. We assume that two conserved aspartates of the Tt72 pol active site, as in other DNA polymerases [64], are involved in the two-metal mechanism that leads to the formation of a phosphodiester bond between the 3′OH end of the primer and the α-phosphate of the incoming nucleotide. A structural and functional analysis of DNA polymerases of all types allowed each subdomain to propose a particular role [65]. The palm subdomain housing the catalytic center also contains the binding site for the 3′-terminus of the primer highlighted by a highly-conserved Tyr residue (Y766, Eco pol I; Y445, Tt72 pol) that participates in forming the NTP-binding site (motif 4). The latter also engages the highly-dynamic fingers subdomain, which, in addition, binds and positions the template strand. In this respect, in the case of Eco pol, an essential role is played by R841 (R569, Tt72 pol), which is required for stabilizing the template strand [66]. In motif 4 of Tt72 pol (Figure 1a), we noted the presence of conserved F441, which in Eco pol I (F762) or Taq pol I (F667) constrains the nucleotide and thus provides discrimination against dideoxynucleotides [27]. In this respect, an important role could also be played by E389 of motif 3 (palm), which in Eco pol I (E710) sterically blocks incoming ribonucleotides, wherein the cellular pool is much greater than that of dNTP [67,68] and prevents their incorporation. Structural and biochemical lines of evidence suggest that both residues are highly-conserved in PolA enzymes and are critical in forming the extremely effective steric gate, excluding nucleotides other than dNTP [69,70].

The significance of conserved residues D384, E389, D615, and S616 in forming the active site of the Tt72 pol nucleotidyltransferase domain was tested using four amino acid substitution variants: D384A, E389A, D615A, and S616A. Individual residues correspond to the position of D705, E710, D882, and E883 of *E. coli* pol I (Figure 1a). To avoid interference, they were constructed using the Tt72 pol gene variant that codes for an enzyme deprived of 3′-5′ exo activity (D78A exo^−^). The double mutants (D78A/D384A, D78A/D389A, D78A/D615A, and D78A/S616A) were purified and assayed for nucleotidyltransferase activity. We found that the activity of two variants was severely impaired compared to the wild-type enzyme (100%) and D78A variant (89 ± 5%): D384A (4 ± 2%) and D615A (4 ± 3%) (Figure 7b). The activities of two other variants (E389A and S616A) were only slightly lowered (35 ± 5% and 76 ± 3%, respectively) (Figure 7b).

The DNA binding capacity of substitution variants was determined using the microscale thermophoresis method (Figure 7b). The obtained K_D_ values for D384A (K_D_ = 100 ± 10 nM), D389A (K_D_ = 68 ± 6 nM), D615A (K_D_ = 78 ± 8 nM), and S616A (K_D_ = 129 ± 1 nM) are close to the paternal D78A exo^−^ variant (K_D_ = 109 ± 5 nM). The results indicate the critical role of conserved acidic residues D384 and D615 in nucleotidyltransferase active site formation. The E389 residue seems to play an accessory role and, as in the case of E710 in Eco pol I, is necessary for forming a stable ternary complex [71].

### 2.7. Characterization of the Tt72 Pol Optima

The recombinant enzyme activity was determined in different settings by measuring the incorporation of [^3^H]-dTTP into activated calf thymus DNA. The effect of temperature was tested in the range of 25–80 °C. It was found that Tt72 pol was active over a wide temperature range, with the highest activity at 55 °C (Figure 8a). This finding is quite surprising, considering that the optimal temperature for *T. thermophilus* MAT72 growth is 65–72 °C, with minimal and maximal temperatures of 47 °C and 85 °C, respectively [72]. 

In the next step, we heated the enzyme for 4 h at 55 °C and, even after an extended period of incubation, we observed high (>80%) Tt72 pol activity (Figure 8b). We also tested the enzyme activity after incubation for 10 min at 55, 60, 65, 75, 85, and 95 °C (Figure 8c). We found that the enzyme retained >80% of its activity after heating at 55 °C and 60 °C, and the enzyme’s activity was residual at higher temperatures (>60 °C). However, during heat treatment (10 min at 75 °C), compatible solutes such as trehalose, trimethylamine *N*-oxide (TMAO), and betaine at a concentration of 0.5 M protected the enzyme against thermal inactivation. We observed that in the case of the first two extremolytes, the enzyme retained >80% of its original activity (Figure 8d). An analysis of the pH effect from 7.0–10.0 revealed an optimum pH at 8.5 (Figure 8e). The following experiment concluded that the Tt72 pol highly depends on Mg^2+^ cations, with the highest activity at 0.5–0.75 mM MgCl_2_ (Figure 8f). In the case of ionic strength, it was found that the highest Tt72 pol activity was observed at a low concentration of KCl and (NH_4_)_2_SO_4_, which corresponds to 25 mM and 10 mM, respectively (Figure 8g,h).

### 2.8. Thermostability of Tt72 DNA Polymerase

Despite the optimum temperature being much lower than expected, the Tt72 pol exhibits the characteristics of proteins derived from thermophiles. The amino acid sequence composition-based r_i_ index for Tt72 pol was calculated to be 6.01, which corresponds with proteins from hyperthermophiles (r > 4.5). This parameter indicates a strong preference of thermophilic organisms for Glu, Lys, Tyr, and Ile over Gln, His, Ala, and Cys, whose high content is noted in proteins from mesophilic sources [73]. Also, the value of CvP bias (charged vs. polar amino acids; [74]) of 12.37 suggests the enzyme’s thermophilic/hyperthermophilic origin. However, this contradicts our finding that the optimal temperature for the enzyme is 55 °C (Figure 8a). It implies that Tt72 pol is a moderately thermostable protein. Therefore, in the next step, the thermal stability of the Tt72 pol was investigated in detail using differential scanning calorimetry (DSC). The analysis of the heat-capacity curve of the protein under study (Figure 9a) indicates that the T_m_ (melting temperature) for the enzyme is 74.6 ± 0.2 °C (ΔH_cal_ = 2.05 × 10^4^ cal mol^−1^). The data presented are mean values from three independent measurements. It was found that there are no significant negative peaks in the heat-capacity curves until t = ~80 °C, which demonstrates that the protein does not aggregate in the range of temperatures between 25–80 °C [75,76]. However, the calorimetric measurements also show that negative peaks appear over t = 80 °C, suggesting that if the temperature of the experimental settings exceeds the value of 80 °C, protein aggregates can be formed. The protein undergoes the unfolding process in the temperature range of 60–76 °C. Above t = ~76 °C, the concentration of the unfolded structure increases rapidly in the solution. For the Tt72 pol, the irreversible process of protein denaturation is observed. It can be concluded that above 60 °C, with increasing temperature, the Tt72 pol slowly starts losing its unique tertiary structure. We also found that T_m_s values of substitution variants concerning residues that are critical for 3′-5′ exo and nucleotidyltransferase domain functioning correspond to the T_m_ of the wild-type enzyme (data not shown). 

We also analyzed the protein thermal stability from the unfolding curves with the help of circular dichroism. The CD spectra were recorded in 20 mM MES (pH 5.0) and 50 mM NaCl in the temperature range of 20–80 °C (Figure 9b). The results obtained indicate that the molar ellipticity varies with temperature. At the lowest temperature selected (t = 20 °C), negative bands at ~208 and ~222 nm, which are characteristic of helical conformation, are most pronounced [77,78,79]. With increasing temperature, the content of the dominant helical form of Tt72 pol decreases in favor of other structures (i.e., turns, beta-sheets, and statistical coils) (Table 1). Starting from t = ~60 °C, we observe the protein unfolding process. The CD spectra confirm the results obtained by the DSC method.

## 3. Discussion

DNA polymerases are remarkable enzymes with the outstanding ability to copy DNA molecules with high accuracy and speed. The present study describes the gene cloning, expression, purification, and characterization of DNA polymerase from phage Tt72 infecting the *T. thermophilus* strain, MAT72. The enzyme belongs to the type-A family of DNA polymerases, which groups replicases involved mainly in DNA repair and processing Okazaki fragments in DNA replication. They are ubiquitous in Bacteria and Eukarya but absent in Archaea [32]. Another unexpected feature associated with DNA polymerases is a low identity level between enzymes derived from phylogenetically close microorganisms. This observation is also the case for Tt72 pol, which shows low similarity to A-family DNA polymerases of *Thermus* phages such as G20c [82], TSP4 [83], P23-45, and P74-26 [84]. However, the Tt72 pol shows significant identity with DNA polymerases of other *Thermus* phages, φTMA [49] and φYS40 [48], which, along with the Tt72 phage, belong to the family Myoviridae. These enzymes share a similar organization with two well-defined domains that are responsible for 3′-5′ exonuclease and nucleotidyltransferase activity. Proteins involved in the processes of the utmost importance for living organisms commonly show a high conservancy level. Replication should obey this rule; however, as mentioned above, DNA polymerases of all types show little similarity. This lack of general conservation suggests convoluted routes of evolution for DNA replication proteins [16]. A phylogenetic analysis of many type-A family DNA polymerases suggests that they originated in bacteria or phages and were spread through horizontal gene transfer between bacteria, viruses, and eukaryotes [15,85]. In this respect, we tested whether Tt72 pol can complement Eco pol I deficiency in the *E. coli polA12*^ts^ strain. The *polA12*^ts^ mutation causes misfolding of the Eco pol I, which decreases 4-fold for nucleotidyltransferase and 5′-3′ exonuclease activity at 42 °C [57]. As a result, we found that the Tt72 pol failed at the complementation assay involving the replication of the *polA*-dependent ColE1-type plasmid replicon. Tt72 pol’s case can be explained by the apparent lack of a 5′-3 exonuclease activity that is essential for removing RNA primers from the 5′-end of the downstream fragments. Consequently, it excludes the Okazaki fragment processing that is critical to DNA synthesis on the lagging strand. However, it was proved that DNA polymerases from microorganisms that are not close phylogenetic relatives of *E. coli* (γ-proteobacteria)—only if they possess 5′-3′ exo activity—can complement a *polA* mutation. In the case of thermophilic enzymes, such property contains the DNA polymerase of *T. aquaticus* [18]. The present work shows that Tth pol I (5′-3′ exo^+^) of *T. thermophilus* can also easily complement *E. coli* Eco pol I deficiency. 

Type-A family enzymes share a similar modular organization and common fold with well-defined domains despite their overall low sequence similarity. In this respect, the Tt72 pol is not different and comprises two distinct structural subunits of approximately 200 and 500 aa in size that contain highly-conserved motifs that are characteristic of the A-family of DNA polymerases [11,56]. A detailed analysis of the molecular model of the Tt72 pol based on crystal structures of DNA polymerases from different sources (PDB 4xvi, 1nzj, 1cmw A, 4x0p, 6vde A, 4ktq A) revealed the spatial architecture of an enzyme with two active sites separated by approx. 30 Å. 

The smaller, globular *N*-terminal domain possesses 3′-5′ exonucleolytic activity, which is involved in removing incorrectly incorporated nucleotides by a proofreading mechanism. The substrate for the reaction carried by 3′-5′ exo is partially single-stranded DNA. The crystal structure analysis of the large Eco pol I (Klenow) fragment, a canonical enzyme for the type-A family, shows that the 3′-5′ active site melts double-stranded DNA (~4–5 nt of the primer strand) and then binds a single-stranded product. In the next step, a misincorporated nucleotide is removed from the primer strand terminus. We hypothesize that in Tt72 pol, the removal reaction follows two ion mechanisms, as in the case of other PolA enzymes [61]. It involves divalent cations (Mg^2+^, Mn^2+^) that are coordinated to the catalytic site by carboxylate groups of four acidic amino acids. In the case of Tt72 pol, those are D15, E17, D78, and D184, which are part of highly-conserved motifs (Exo I, Exo II, and Exo III). The deprotonation of a water molecule at the catalytic center initiates hydroxide ion nucleophilic attack on the phosphodiester bond of the misincorporated nucleotide. While testing Tt72 pol, we found that the enzyme exonucleolytic activity is observed even in the presence of 0.4 mM dNTP. In general, at higher dNTP concentrations, DNA polymerases’ exonucleolytic activity is suppressed [86]. Thus, our results prove that Tt72 pol possesses a strong proofreading activity. Strong 3′-5′ exo activity was also observed in the *T. thermophilus* phage YS40 DNA polymerase, which is homologous to Tt72 pol [48].

The nucleotidyltransferase domain of type-A family enzymes can be divided into three subdomains: palm, fingers, and thumb. They contain six evolutionary, highly-conserved amino acid motifs. The palm subdomain with the catalytic site, located at some distance from the 3′-5′ exo active site, participates in the nucleotidyl transfer reaction. In this respect, in the structure of Tt72 pol, we have pointed out two acidic residues (D384 and D615). We assume that, similar to other A-family enzymes [64,87], both residues coordinate two divalent metal ions that are engaged in catalysis. The third acidic residue, E389, seems to have an accessory role in forming the enzyme-template ternary complex. The highly-conserved active site geometry suggests a universal strategy for the metal-promoted polymerization reaction that builds DNA strands. According to the proposed mechanism, one metal ion promotes the deprotonation of O3′ of the primer strand, leading to the formation of the attacking oxyanion (O3′^−^). This step is followed by the nucleophilic attack (S_N_2-type reaction) by the O3′^−^oxyanion on the α-phosphate of the incoming 2′-deoxynucleoside triphosphate. The second ion facilitates the formation of the pentavalent transition state at the phosphate in the position α of the dNTP and the release of the pyrophosphate that follows the incorporation of a nucleoside monophosphate by the formation of a phosphodiester bond. Pyrophosphate, a high-energy intermediate, is further hydrolyzed into two phosphate molecules in a reaction that is catalyzed by the DNA polymerase itself. This step ensures that the free energy change (ΔG) on the DNA synthesis reaction is negative so that it can proceed spontaneously in the forward direction [88]. Many crystal structures of DNA polymerases support the two-metal-ion mechanism. However, recently, this classical model has been challenged by a three-metal-ion mechanism that imposes the involvement of an additional divalent metal ion, as reported for type-Y Pol η, type-X family Pol β, and Pol µ [89,90,91,92]. It was found that a third metal is transiently coordinated upon binding of correct, but not incorrect, nucleotides. This mechanism seems to be adopted by enzymes that are mainly involved in DNA repair. The role of the third ion in catalysis is still a matter of dispute [93]. Also, a recent, detailed analysis of high-resolution crystal structures of PolA-type *Geobacillus stearothermophilus* DNA polymerase I intermediates allowed the capture of new conformations for the enzyme translocation and nucleotide pre-insertion step that underlies the conformational dynamic nature of enzymatic DNA synthesis [94]. Such dynamics, which are highlighted for the fingers subdomain, are associated with changes from an open to a closed state that allows the enzyme active site to be positioned for nucleotide incorporation.

Our results revealed that Tt72 pol is moderately thermostable when tested in vitro (T_m_ = 74.6 ± 0.2 °C). We observed similar moderate thermoresistance in the case of Tt72 RecA protein [42]. These findings were entirely unexpected as proteins produced by thermophilic microorganisms usually are more resistant to thermal denaturation than their mesophilic counterparts [95]. It is also surprising considering the optimal growth temperature of the host bacterium, *T. thermophilus,* with an upper limit at 85 °C. We hypothesize that the enzyme can have much higher thermal stability inside the host cell due to interactions with other proteins and compatible solutes; the last refers to compounds like sugars, polyols, amino acids, ectoines, and betaines that accumulated inside the cell without affecting its viability [96]. Indeed, we found that during heat treatment (10 min at 75 °C), trehalose, trimethylamine *N*-oxide, and betaine protected the enzyme against thermal inactivation. These chemical chaperones are polar, highly-soluble molecules, and are uncharged at physiological pH [97,98]. Early observations of their effect on proteins formulated a hypothesis that compatible solutes, due to their thermostabilizing properties, are part of a survival strategy that adapts thermophiles to life at high temperatures [99,100]. Other extremophilic microbes use this strategy as a universal route for intrinsic physiological adjustments concerning the supraoptimal environmental conditions [101,102,103]. For example, *T. thermophilus* growth above the optimum temperature results in trehalose accumulation and 2-*O*-β-mannosylglycerate, both considered thermostabilizing compounds [104,105,106]. The stabilizing potential of compatible solutes was shown for DNA polymerases from *T. antranikianii* and *T. brockianus* [26]. After 15 min of incubation at 80 °C, it was found that both enzymes retained only 10% of the activity compared to the unheated sample. Adding L-proline to 0.5 M increased the enzymes’ thermal stability, as about 50% of their activity remained after heat treatment. At the molecular level, compatible solutes as highly hydrophilic compounds are preferentially excluded from the hydration shell of native proteins. This steric exclusion promotes the stabilization of the secondary and tertiary protein structures, thus resulting in their higher thermoresistance [107,108]. In this respect, hydrogen bonds are essential to contributing positively to protein stability [109]. A recent report showed that glycerol, sorbitol, and glucose effectively shorten protein hydrogen bonds through a mechanism that involves competition with a solvent. Compatible solutes stabilize protein folding by weakening protein–solvent hydrogen bonds, which in turn strengthens hydrogen bonds within the protein molecule [110]. 

For phages, as for any other organism, the most critical part of the life cycle is the stage of copying the genetic material to be passed to the progeny. The PolA-type DNA polymerase that is characterized in the present work is a key element of the *T. thermophilus* Tt72 phage molecular nanomachine, which is responsible for viral DNA replication. In a broader sense, our research is directed to define the Tt72 phage replisome constituents. We expect other proteins to be involved as each replicon uses its own logic and economy of genome duplication [19,31,111]. Some of them may be harnessed from the replication machinery of the native host bacterium. So far, we have identified genes coding for primase and a DnaB-like helicase in the Tt72 phage genome. However, we hope to find other genes that are involved in replication, as thermophilic phages’ genetic blueprints remain largely unexplored. In the next step, we will focus on characterizing respective proteins functions to get a close insight into the Tt72 phage replication cycle and its molecular aspects. 

## 4. Materials and Methods

### 4.1. Bioinformatics Analysis

A sequence-similarity search was performed using BlastP (https://blast.ncbi.nlm.nih.gov/blast.cgi; accessed on 18 March 2022) against a non-redundant database using the default settings. The amino acid sequences were also analyzed using Pfam [112] and the Conserved Domain Database [113] to find annotations across all protein families. Protein sequences were aligned using the CLUSTAL W program [114]. The three-dimensional structure of Tt72 pol was predicted using homology modeling. A homology model was built using Phyre2 (intensive mode) using multiple templates (PDB: 4xvi_A, 1njz_A, 1cmw_A, 4x0p_B, 6vde_A, 4ktq_A) [115], and then side chains were refined by GalaxyRefine [116]. The model was visualized and interpreted using USCF Chimera [117]. The codon adaptation index was calculated using the JCat computer program [52]. Gene-encoding Tt72 pol was analyzed with DNASIS software (Hitachi Software Engineering), and the molecular weight and isoelectric point (pI) were predicted by the IPC 2.0 webserver [118]. The index r_i_ and the CvP bias parameters were calculated as reported [73,74]. The files related to the bioinformatics analysis are available at: https://doi.org/10.18150/0D4BNT (accessed on 18 March 2022).

### 4.2. Bacterial Strains, Plasmids, and Phage

The vB_Tt72 phage and the host strain, *T. thermophilus* MAT72, were isolated from the hot spring in the Badstofuhver area in Hveragerdi, Iceland, and obtained from the MATIS collection of microorganisms, Iceland. *T. thermophilus* MAT72 cells were grown at 60 °C in TM medium [47], whereas *E. coli* strain DH5α was used for molecular cloning experiments. Next, *E. coli* strains JS200 *recA718 polA12*^ts^ [57] and Bl21(DE3)[RARE] were used for genetic complementation assay and protein overproduction, respectively. *T. aquaticus* YT1, used as a source of a *polA* gene, was obtained from Dr. Roger K. Latta (National Research Council, Canada). *E. coli* strains were cultivated in Luria broth (LB) or Luria agar (LA) solid medium [119] and supplemented with ampicillin (Ap, 100 µg/mL), chloramphenicol (Cm, 34 µg/mL), kanamycin (Km, 30 µg/mL), or tetracycline (Tc, 15 µg/mL) when necessary. For the [^35^S]-methionine incorporation experiment, bacteria were cultivated in an M-9 minimal medium [119]. Phage vB_Tt72 genomic DNA was isolated using a standard protocol [119]. The following vectors were used in this work: pET15b (Ap^R^), pET24a(+)(Km^R^) (Novagen, Madison, WI, USA), and pBR322 (Ap^R^, Tc^R^) with a *polA*-dependent pMB1 origin of replication. The low-copy-number vector pHSG576 (Cm^R^) with a *polA*-independent pSC101 origin of replication [120] and its derivative pHSG_pol_I_wt encoding Eco pol I were used in a genetic complementation assay [57]. Both plasmids were obtained from Dr. Manel Camps (University of California Santa Cruz, USA). The plasmids constructed in this work were deposited in the Collection of Plasmids and Microorganisms at the University of Gdansk, Gdansk, Poland.

### 4.3. Cloning of Tt72 polA Gene

Standard protocols were used for molecular cloning [119]. Restriction fragments for cloning were extracted from agarose gels using a DEAE-cellulose membrane method [121]. For enzyme overproduction, two expression plasmids were constructed using phage Tt72 genomic DNA as a template for PCR-based DNA amplification with Q5 High-Fidelity DNA polymerase (New England Biolabs, Ipswich, MA, USA). Each of them encodes a recombinant Tt72 pol with a His-Tag at the *N*-(pET15b_polTt72) or *C*-terminus (pET24a(+)_polTt72). For the first plasmid construction, the gene coding for Tt72 pol was amplified with primers polTt72-NdeI-F and polTt72-BamHI-R (Appendix A). After digestion with NdeI and BamHI restriction enzymes, the PCR product (2.1-kb) was ligated into the expression vector, pET15b. In the case of the second plasmid, the DNA fragment (2.1-kb) amplified with primers polTt72-NdeI-F and polTt72-SalI-R (Appendix A) was digested with NdeI and SalI and cloned into pET24a(+). For testing genetic complementation, the gene coding for Tt72 pol was amplified with primers polTt72-PstI-F and polTt72-BamHI-R (Appendix A). The resulting DNA fragment (2.1-kb), after processing with PstI and BamHI, was cloned into a pHSG576 vector, which gave pHSG_polTt72. Similarly, the genes coding for DNA polymerases from *T. thermophilus* MAT72 (Tth pol I) and *T. aquaticus* YT1 (Taq pol I) were amplified with primer pairs polTth-Sal-F, polTth-EcoRI-R, polTaq-SalI-F, and polTaq-EcoRI-R (Appendix A), respectively. The resulting PCR products (2.5-kb each) were digested with SalI and EcoRI and cloned into pHSG576, giving plasmids pHSG_polTth and pHSG_polTaq, respectively. Amplification reactions were carried out using chromosomal DNA that was isolated from *T. thermophilus* MAT72 or *T. aquaticus* YT1 as templates. Site-directed mutagenesis was used to introduce missense mutations in the codons of catalytic amino acids of the exo 3′-5′ domain (Asp15, Glu17, Leu27, Asp78, Tyr180, and Asp184) and the Tt72 pol nucleotidyltransferase domain (Asp384, Glu389, Asp615, and Ser616). The protocol followed the instructions of the QuickChange II Site-Directed Mutagenesis Kit (Agilent Technologies, Santa Clara, CA, USA). The oligonucleotides used as primers in the mutagenesis procedure are listed in Appendix A. pEC15b was used as a vector plasmid. All recombinant clones were verified by DNA sequencing. 

### 4.4. Genetic Complementation

*E. coli* JS200 *recA*718 *polA*12^ts^ [pBR322] cells were transformed with plasmid pHSG576 (Cm^R^, Eco pol I-independent pSC101 origin of replication; [120]) and its derivatives carrying respective *polA* genes: pHSG_pol_I (Eco pol I), pHSG_polTt72 (Tt72 pol), pHSG_polTth (Tth pol), or pHSG_polTaq (Taq pol). The transformed cells were cultivated at 30 °C in LB medium supplemented with Ap, Cm, and Tc to an A_600_ of 0.5. To determine complementation efficiency of Tt72 pol or bacterial enzymes, Eco pol I, Tth pol I, and Taq pol I, and their respective cultures, were serially diluted 10-fold with LB medium, and 5 µL of each dilution was spotted on LA plates supplemented with Ap, Cm, Tc, and isopropyl-β-D-thiogalactopyranoside (IPTG, 1 mM). Duplicate plates were incubated for 48 h at 30 °C or 42 °C.

### 4.5. Overexpression of the Tt72 polA Gene

To estimate the rates of protein synthesis in *E. coli* Bl21(DE3)[pRARE][pET15b_polTt72], pulse-labeling with [^35^S]-methionine was performed. The cells were grown at 30 °C in an M9 medium supplemented with Ap and Cm to A_600_ of 0.5. At this point, the overproduction of Tt72 pol was induced by adding IPTG to the final concentration of 1 mM. The *E. coli* RNA polymerase inhibitor rifampicin was added 30 min after induction to a concentration of 200 µg/mL. A hundred and fifty µL samples of culture were labeled with 2.5 µCi [^35^S]-methionine (1175 Ci/mmol; PerkinElmer, Waltham, MA, USA) for 5 min at 30 °C in 30 min intervals. Cells were centrifuged and resuspended in SDS/PAGE loading dye (125 mM Tris-HCl (pH 6.8), 5% [*v*/*v*] SDS, 10% [*v*/*v*] 2-mercaptoethanol, 10% [*v*/*v*] glycerol and 0.1% [*w*/*v*] bromophenol blue). Then, samples were analyzed on 10% SDS-PAGE and transferred to a polyvinylidene difluoride (PVDF) membrane (GE Healthcare Life Sciences, Uppsala, Sweden). Protein bands were detected by autoradiography.

### 4.6. Purification of Tt72 DNA Polymerase

*E. coli* Bl21(DE3)[pRARE] cells harboring pET15_polTt72 were cultivated at 30 °C in an LB medium supplemented with Ap and Cm to an A_600_ of 0.5. The temperature was reduced to 18 °C and the Tt72 pol overproduction was induced by adding IPTG to 1 mM. The culture was grown for an additional 16 h at 18 °C, and the cells were harvested by centrifugation and stored at −80 °C. The His-tagged recombinant enzyme was purified using affinity chromatography and the ÄKTA Pure chromatography system (GE Healthcare). All purification procedures were carried out at 4 °C. For the standard purification procedure, 4 g of cell paste was resuspended in 6 mL buffer A (50 mM NaH_2_PO_4_ (pH 8.0), 500 mM NaCl, 0.1% [*v*/*v*] Triton X-100, 10% [*v*/*v*] glycerol) containing 10 mM imidazole. The cells were disrupted by sonication (4 °C, 30 bursts of 10 s at an amplitude of 12 µm; MISONIX sonicator XL2020; Misonix Inc., Farmingdale, NY, USA), and the resulting lysate was clarified by centrifugation (4 °C; 10,000× *g*; 20 min). In the next step, the lysate was incubated for 20 min at 60 °C to remove host proteins. The precipitated proteins were discarded by centrifugation (4 °C; 10,000× *g*; 20 min). The clarified lysate was loaded onto a HiTrap^TM^ TALON column (GE Healthcare) equilibrated with buffer A. The column was washed with buffer A and the bound protein was eluted in the same buffer supplemented with 200 mM imidazole. Peak fractions containing substantial quantities of the target protein were pooled and dialyzed against buffer B (20 mM Tris-HCl (pH 8.0), 50 mM NaCl and 5% [*v*/*v*] glycerol). Then, proteins were loaded onto the HiTrap^TM^ Heparin HP column (GE Healthcare) equilibrated with buffer B. The column was washed with the same buffer. Bound proteins were eluted with a 50 to 1000 mM NaCl gradient in buffer B. Fractions containing the highest activity of Tt72 pol were pooled and dialyzed against 20 mM Tris-HCl (pH 8.0), 50 mM NaCl, 1 mM DTT, 0.1 mM EDTA, and 50% [*v*/*v*] glycerol, and then stored at −20 °C. The protein concentration was determined by the Bradford assay [122]. The purity of Tt72 pol was monitored by SDS-PAGE and by size exclusion chromatography on a Superdex 75 10/300 GL column using the ÄKTA Pure 25 system (GE Healthcare). The same procedure has been used to purify Tt72 pol substitution variants. 

### 4.7. Polymerase Activity Assay

The incorporation of radiolabeled nucleotides was used to assay the polymerase activity of the Tt72 pol. The reaction mixture (50 µL) consisted of Tt72 buffer (10 mM Tris-HCl (pH 8.5), 25 mM KCl, 0.5 mM MgCl_2_, 10 mM (NH_4_)_2_SO_4_) supplemented with 200 µM of dATP, 200 µM dCTT, 200 µM dGTP, 20 µM dTTP, 3 µCi/mL of [methyl-^3^H]-thymidine 5′-triphosphate (84.2 Ci/mmol; PerkinElmer), and 0.2 mg/mL of activated calf thymus DNA. The reactions started after adding aliquots of the Tt72 pol on ice; samples were immediately transferred to a preheated thermal cycler and incubated for 10 min at 55 °C, then stopped by adding 200 µL of 10% [*w*/*v*] trichloroacetic acid (TCA) and kept on ice for 10 min. After termination, an aliquot was spotted on a GF/C filter disc (Whatman, Clifton, NJ, USA). The filters were washed four times with 5 mL of 5% [*w*/*v*] TCA, then washed twice with 5 mL of 70% [*v*/*v*] ethanol. The filters were air-dried and counted in a liquid scintillation counter (PerkinElmer). One unit of the enzyme was defined as the amount of enzyme that incorporated 10 nmol of dNTPs into acid-insoluble material in 30 min at 55 °C. 

### 4.8. 3′-5′ Exonuclease Assay

To prepare the 3′ end-labeled DNA substrate, sticky ends of DNA restriction fragments (λ/HindIII) were filled in a reaction mixture containing 50 mM Tris-HCl (pH 8.0), 5 mM MgCl_2_, 1 mM DTT, 100 µM dATP, 100 µM dGTP, 100 µM dCTP, 1 µCi [methyl-^3^H]-thymidine 5′-triphosphate (60–70 Ci/mmol; PerkinElmer), 4 µg λ/HindIII, and 1 U of Klenow Fragment (Thermo Scientific). After labeling (15 min, 37 °C), DNA fragments were purified by a Clean-up AX kit (A&A Biotechnology, Gdansk, Poland). The 3′-5′ exonuclease activity of the Tt72 pol was assayed in a 50 µL reaction mixture containing 10 mM Tris-HCl (pH 8.5), 25 mM NaCl, 0.5 mM MgCl_2_, 10 mM (NH_4_)_2_SO_4,_ and 0.5 µg of the labeled substrate at 55 °C for 10 min in the presence or absence of dNTPs. The reaction was stopped by cooling samples on ice and adding 200 µL of 10% [*w*/*v*] TCA. After termination, an aliquot was spotted at a GF/C filter disc (Whatman). The filters were washed four times with 5 mL of 5% [*w*/*v*] TCA, then washed twice with 5 mL of 70% [*v*/*v*] ethanol. The filters were air-dried and counted in a liquid scintillation counter (Perkin Elmer). This allowed us to calculate the percentage of radioactivity released from labelled DNA fragments.

### 4.9. Terminal Transferase Assay

Oligonucleotides Tt72-Cy3 (24-mer) and Tt72-blunt (24-mer) were annealed at a 1:1 molar ratio (Appendix A) to give a blunt-ended double-stranded substrate. The reaction mixture (5 µL) contained 0.5 pmol of the Cy3-labeled 24-bp double-stranded oligonucleotide in Tt72 buffer and was supplemented with dNTPs brought to 0.4 mM and 100 nM Tt72 pol. The samples were incubated for 20 min at 55 °C. The reaction was stopped by adding 5 µL of formamide to each tube, followed by denaturation at 95 °C for 10 min. The quenched reaction samples were separated on a 15% polyacrylamide denaturing gel containing 8 M urea in TBE buffer (89 mM Tris base, 89 mM boric acid, 2 mM EDTA-Na_2_, (pH 8.0)) and scanned using a Typhoon 9200 phosphorimager (GE Healthcare).

### 4.10. Western Blotting

Purified protein samples were separated by electrophoresis (10% SDS-PAGE) and electroblotted onto a PVDF membrane using a dry transfer system (2 h, 270 mA). Subsequently, the membrane was blocked in the PBS-T buffer (137 mM NaCl, 2.7 mM KCl, 10 mM Na_2_HPO_4_, 1.8 mM KH_2_PO_4_, 0.05% [*v*/*v*] Tween 20, (pH 7.4)) with 3% skimmed milk for 1 h at room temperature. Next, the membrane was incubated overnight at 4 °C with a specific primary antibody (Anti-His-tag antibody 1:1000, Sigma-Aldrich, Cat No. SAB1306082) in PBS buffer (137 mM NaCl, 2.7 mM KCl, 10 mM Na_2_HPO_4_, 1.8 mM KH_2_PO_4_, (pH 7.4)) with 3% skimmed milk. After three washing cycles with PBS-T with 3% skimmed milk, the membrane was incubated with the secondary antibody (anti-rabbit antibody conjugated with alkaline phosphatase, 1:20,000, Sigma-Aldrich, Cat No. A3687) for 1 h at room temperature. Finally, the membrane was washed three times in PBS-T buffer for 10 min at room temperature for each cycle. X-ray film and Pierce^TM^ ECL Plus Western Blotting Substrate (Thermo Scientific, Cat No. 32134) have been used to detect a Western blot chemiluminescent signal.

### 4.11. Analytical Ultracentrifugation

Sedimentation velocity experiments were performed using a ProteomeLab XL-I analytical ultracentrifuge (Beckman-Coulter Inc., Brea, CA, USA) equipped with an An-60, 4-hole analytical rotor, and double-sector charcoal-Epon cells (12-mm path length). The experiments were carried out at 4 °C at 50,000 rpm, using the continuous scan mode and a radial spacing of 0.003 cm. Cells were loaded with 400 μL of sample and 410 μL of buffer (20 mM Tris-HCl (pH 8.0), 50 mM NaCl, and 5% [*v*/*v*] glycerol). After loading into the centrifuge and reaching a temperature of 4 °C, the cells were equilibrated for 1 h. Scans were collected at 6 min intervals between measurements in absorbance mode at 280 nm. Buffer density (d = 1.019 g/mL) and viscosity (viscosity = 1.895 mPas) were measured at 4 °C using an Anton Paar (Graz, Austria) DMA 5000 densitometer and Lovis 2000 M rolling-ball viscometer. The partial specific volume, v-bar = 7382 g/mL, and extinction coefficient for protein were calculated using SEDNTERP software version 1.09 (Informer Technologies Inc., Dallas, TX, USA; http://bitcwiki.sr.unh.edu, accessed on 18 March 2022). Data were analyzed using the “Continuous c(s) distribution” model of the SEDFIT (version 16.1c) program. The confidence level (F-ratio) was specified as 0.6.

### 4.12. Differential Scanning Calorimetry

Calorimetric measurements were made with a VP-DSC microcalorimeter (MicroCal Inc., Northhampton, MA, USA) at a scanning rate of 90 °C /1 h. Scans were obtained at a protein concentration of 9.1 µM. The cell volume was 0.5 mL. All scans were run in 20 mM Tris-HCl (pH 8.0) and 50 mM NaCl in a temperature range from 25 °C to 90 °C. The reversibility of the transition was checked by cooling and reheating the same sample. These measurements were recorded three times. Differential Scanning Calorimetry (DSC) measurements were analyzed using Origin 7.0 software from MicroCal [123]. The quantity measured by DSC is the difference between the heat capacity of the Tris buffer–protein solution and that of pure Tris buffer. During DSC measurements, heat absorption causes a temperature difference (T) between the cells when a protein unfolds. The reference cell was filled with buffer and the sample cell with the protein solutions. They were then heated at a constant scan rate (90 °C /1 h). The “none” type of feedback model was chosen for all DSC experiments.

### 4.13. Circular Dichroism Spectra

Circular dichroism (CD) spectra were recorded in the water on a Jasco-715 automatic recording spectropolarimeter (Jasco Corporation, Tokyo, Japan) for Tt72 pol in 20 mM MES (pH 5.0) and 50 mM NaCl. The experiments were performed in the range of 20–80 °C every 10 degrees. Measurements were made in 1 mm quartz cuvettes. The spectra were recorded in the 200–260 nm wavelength range, using a sensitivity of five millidegrees and a scan speed of 50 nm/min. The results were plotted as the mean residue ellipticity Θ [degree × cm^2^ × dmol^−1^]. CD measurements were made at a protein concentration of 3.6 µM in 20 mM MES (pH 5.0) and 50 mM NaCl. The volume of samples was 0.3 mL. The secondary structure content was estimated from all the CD spectra using the CONTIN/LL program, a variant of the CONTIN method provided within the CDPro software package [81].

### 4.14. Microscale Thermophoresis

Microscale thermophoresis (MST) was used to analyze the binding affinity between purified Tt72 pol, and its substitutive variants, and partially double-stranded primer/template oligo with a fluorescent label. Assays were performed with the Monolith NT.115^Pico^ instrument (Nano Temper Technologies, Munich, Germany). The substrate oligo was assembled by annealing polTt72-Cy5 (24-mer, primer strand) and polTt72-long (30-mer, template strand) (Appendix A) at a 1:1 molar ratio in a buffer containing 10 mM Tris-HCl (pH 8.0) and 50 mM NaCl. After heating to 95 °C for 5 min, the sample was cooled to 20 °C at 1 °C/min. Measurements were performed at 25 °C in 10 mM Tris-HCl pH 8.5 buffer, 0.5 mM MgSO_4_, 50 mM KCl, 10 mM (NH_4_)_2_SO_4_, and 0.05% Tween 20 at a volume of 20 µL. Reactions were carried out with a constant amount of substrate oligo (1 nM), which was titrated against a decreasing concentration of Tt72 pol or its substitution variants. After incubation for 10 min at 55 °C, the standard capillaries NT.115 (NanoTemper Technologies) were filled with the reaction mixture. The infrared laser power was 40%, and 20% LED power was used. Data from a minimum of three replicate binding assays were analyzed, and the K_D_ was determined by non-linear fitting of the thermophoresis responses using the NTAnalysis software (Nano Temper Technologies).

### 4.15. Nucleotide Sequence Accession Number

The nucleotide sequence of Tt72 pol was deposited to GenBank under the accession number ON714139.

## Figures and Tables

**Figure 1 ijms-23-07945-f001:**
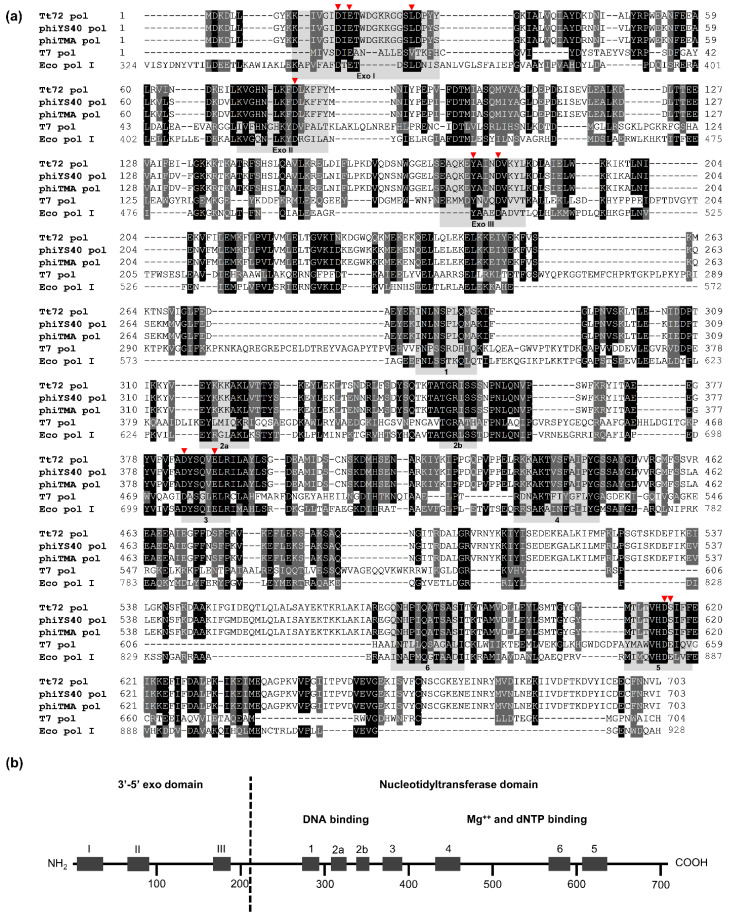
(**a**) Multiple sequence alignment of the Tt72 pol and its type-A family homologs. The alignment was performed using CLUSTAL W. The GenBank accession numbers for the genes coding for Tt72, φYS40, φTMA, T7, and *E. coli* (Pol I) DNA polymerases ON714139, YP_874046.1, YP_004782240.1, NP_041982.1, WP_000250029.1, respectively. Gray shading reflects amino acid conservation at 70% consensus, whereas the black boxes represent 100% amino acid sequence identity. The highly-conserved amino acid motifs within the 3′-5′ exo (Exo I, Exo II, and Exo III) and the nucleotidyltransferase domain (1, 2a, 2b, 3, 4, 5, and 6) are indicated. (**b**) Modular organization of Tt72 pol. The product *polA* gene of phage vB_Tt72 is a protein composed of a 703 amino acid with a 3′-5′ exo and nucleotidyltransferase domain. The positions of motifs characteristic for both domains are indicated. The red triangle indicates the catalytic residues within the 3′-5′ exo and nucleotidyltransferase domain, wherein the importance was verified using site-directed mutagenesis.

**Figure 2 ijms-23-07945-f002:**
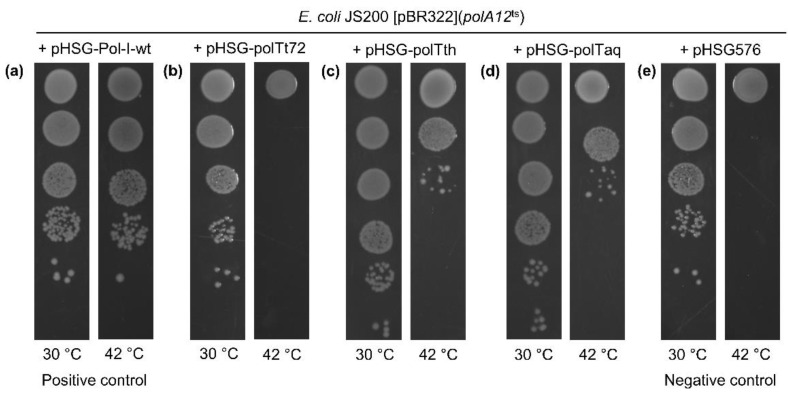
Functional complementation of *E. coli polA12*^ts^ by (**a**) wild-type *E. coli* pol I (positive control), (**b**) Tt72 pol, (**c**) *T. thermophilus* MAT72 pol I, (**d**) *T. aquaticus* pol I, and (**e**) vector pHSG576 (negative control). *E. coli polA12*^ts^ JS200 [pBR322] was transformed with pHSG-Pol-I-wt (Eco pol I), pHSG-polTt72 (Tt72 pol), pHSG-polTth (Tth pol), pHSG-polTaq (Taq pol), and plasmid pHSG576 (low-copy plasmid with a pol I-independent pSC101 origin replication). Transformed cells were grown to A600 of 0.5 in LB medium containing tetracycline (15 µg/mL), chloramphenicol (34 µg/mL), and ampicillin (100 µg/mL). Then, the cells were serially 10-fold diluted with LB medium, and 5 µL of each dilution was spotted on LA plates supplemented with the same antibiotics and IPTG at a concentration of 1 mM. Plates in duplicate were incubated at 30 °C and 42 °C for 24–30 h.

**Figure 3 ijms-23-07945-f003:**
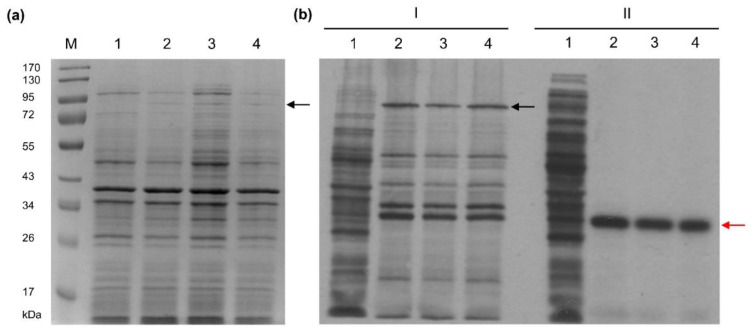
Overproduction of the Tt72 pol in *E. coli* BL21(DE3)[pRARE, pET15b_polTt72] cultivated at 30 °C. (**a**) SDS-PAGE (10%) of cellular proteins from 150 µL samples of *E. coli* BL21(DE3)[pRARE, pET15b_polTt72]: lane 1, before induction; lanes 2–4, after induction (60, 90, and 120 min respectively); 1 mM IPTG; lane M, protein molecular mass markers (PageRuler^TM^ Prestained Protein Ladder, Thermo Scientific). The gel was stained with Coomassie Brilliant Blue R-250. (**b**) Autoradiogram of proteins pulse-labeled in vivo with [^35^S]-methionine and separated on SDS-PAGE (10%). A hundred and fifty µL samples of cultures were labeled for 5 min with 2.5 µCi [^35^S]-methionine at 30 °C. (I) *E. coli* BL21(DE3)[pRARE, pET15b_polTt72]: lane 1, before induction; lanes 2–4, after induction (1 mM IPTG): 2 (60 min), 3 (90 min), and 4 (120 min). Rifampicin was added to a 200 µg/mL concentration 30 min after induction. (II) *E. coli* BL21(DE3)[pRARE, pET15b] (negative control): lanes 1–4 correspond to the treatment described in subpanel I. The Tt72 pol and β-lactamase positions are indicated by the black and red arrows, respectively.

**Figure 4 ijms-23-07945-f004:**
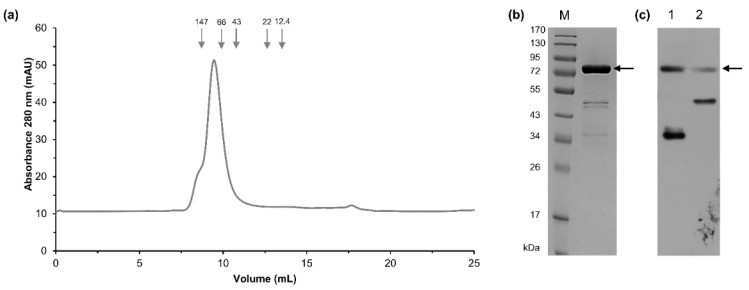
Analysis of Tt72 pol degradation products. (**a**) Size-exclusion chromatography elution profile of Tt72 pol on a Superdex 75 10/300 GL column (GE Healthcare). Tt72 pol (0.5 mg) sample was loaded on the column equilibrated with 50 mM potassium phosphate buffer (pH 7.0), 150 mM NaCl, and eluted at a flow rate of 0.5 mL/min in the same buffer. The column was calibrated with proteins of known molecular masses: alcohol dehydrogenase (tetramer), 146.8 kDa; bovine serum albumin, 66 kDa; ovalbumin, 43 kDa; trypsin inhibitor, 22 kDa; and cytochrome C, 12.4 kDa (Sigma-Aldrich, St. Louis, MO, USA). (**b**) Eluted protein with His-tag at the *N*-terminus was analyzed by 10% SDS-polyacrylamide gel electrophoresis and stained with Coomassie Brilliant Blue. Lane M, protein molecular mass standards (PageRuler^TM^ Prestained Protein Ladder, Thermo Scientific, Waltham, MA, USA). (**c**) Western blot analysis of the Tt72 pol with use of anti-His-tag antibodies. Lane 1, recombinant Tt72 pol with His-tag at *N*-terminus; lane 2, recombinant Tt72 pol with His-tag at *C*-terminus. An arrow indicates the position of the intact His-tagged Tt72 pol (82.7 kDa).

**Figure 5 ijms-23-07945-f005:**
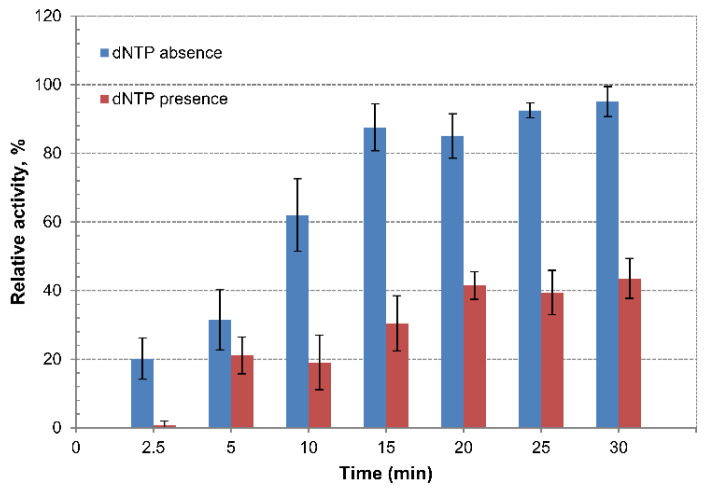
Examination of 3′-5′ exonuclease activity of Tt72 pol. The activity was measured in the absence and presence of dNTPs (100 µM) in the reaction mixture. The reaction was carried out at 55 °C. Values represent the mean ± standard deviation (*n* = 3).

**Figure 6 ijms-23-07945-f006:**
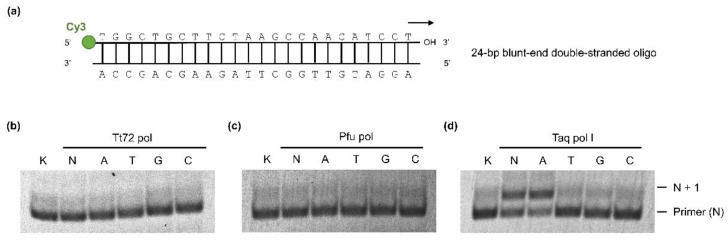
Determination of terminal transferase activity of Tt72 pol. The 24-bp blunt-end double-stranded oligo with the fluorescent label was used as a template (**a**). The products of extension reactions are shown in (**b**) Tt72 pol, (**c**) Pfu pol (negative control), and (**d**) Taq pol I (positive control). The strand extension reaction for each enzyme was carried out for 20 min with either a mixture of all dNTPs or only one: lane A, dATP; T, dTTP; G, dGTP; and C, dCTP, respectively.

**Figure 7 ijms-23-07945-f007:**
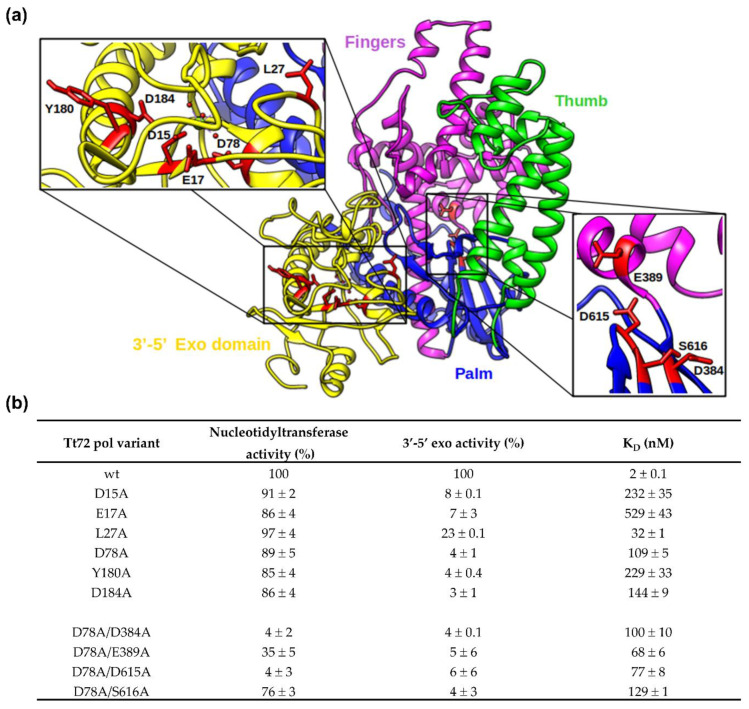
Structural model of Tt72 DNA polymerase (**a**). The 3′-5′ exonuclease domain (aa 1-201) is colored yellow. The nucleotidyltransferase domain (aa 202-703) is divided into palm (residues 202-229, 337-365, 376-386, 606-703; colored blue), thumb (residues 230-336, 366-375; green), and fingers (residues 387-605, magenta) subdomains. Additionally, the residues that are crucial for catalytic activity have been zoomed in (D15, E17, D78, D184, Y180 are responsible for exonucleolytic activity—left, and D384, E389, D615, S616 are involved in the template-directed polymerization of dNTPs onto the growing primer strand of duplex DNA—right). (**b**) Activity and DNA binding affinity (K_D_) of Tt72 pol and its substitution variants.

**Figure 8 ijms-23-07945-f008:**
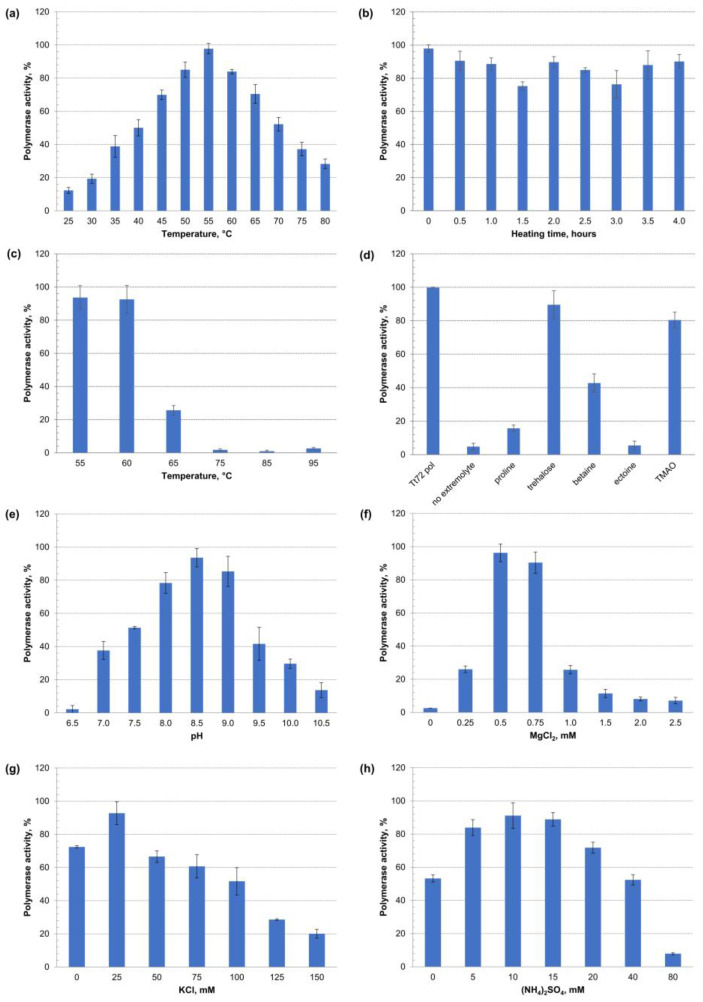
Characterization of Tt72 pol activity optima. Enzyme activity was estimated by measuring the incorporation of radiolabeled nucleotide [^3^H]-dTTP into activated calf thymus DNA at 55 °C; identical aliquots of Tt72 pol were assayed in three independent sample replicates for each experiment. The variable factors were (**a**) temperature (25–80 °C), (**b**) heating time at 55 °C (0–4 h), (**c**) incubation for 10 min at 55–95 °C, and (**d**) heat shock for 10 min at 75 °C in the presence of compatible solutes, such as proline, trehalose, betaine, ectoine, and TMAO at a concentration of 0.5 M. (**e**) pH range (6.5–10.5), (**f**) MgCl_2_ concentration (0–2.5 mM), (**g**) KCl concentration (0–150 mM), and (**h**) (NH_4_)_2_SO_4_ concentration (0–80 mM). Y-axis shows the percent of polymerase activity to maximal activity. Error bars represent SD from the mean.

**Figure 9 ijms-23-07945-f009:**
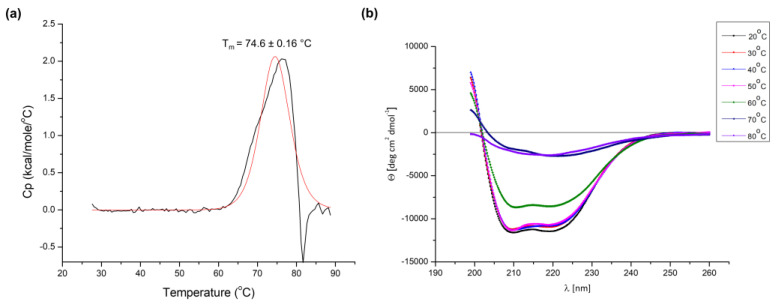
Examination of the thermal stability of the Tt72 pol. (**a**) Tt72 pol’s heat-capacity curves were determined by differential scanning calorimetry (DSC) that was recorded in 20 mM Tris-HCl (pH 8.0) and 50 mM NaCl. The Tt72 pol (black line) and the fit to the two-state model (red line). (**b**) Changes in circular dichroism spectra during thermal denaturation of Tt72 pol. The CD spectra were recorded in 20 mM MES (pH 5.0) and 50 mM NaCl from 200 to 260 nm over a temperature range of 20–80 °C. Representative spectra are shown at specific temperatures during thermal denaturation.

**Table 1 ijms-23-07945-t001:** Predicted changes in the secondary structure of Tt72 pol during thermal denaturation. Changes were calculated by measuring the UV CD spectra from 190 to 260 nM during heating and using the CONTIN/LL [80,81] algorithm provided within the CDPro software package. In CONTIN/LL, the proteins in the reference set are arranged in the order of the increasing RMS distance of the CD spectra from that of the protein analyzed, and the more distant proteins are deleted systematically to construct smaller reference sets. This arrangement results in a set of solutions, one for each LL combination, the number of which is determined by the number of reference proteins and the minimum number of proteins used for a solution. The reference PROTEIN Set Selected was SMP56 (56 Proteins): 43 Soluble + 13 Membrane (with known structures) implemented in the program.

Temperature (°C)	α-Helix [%]	β-Sheet [%]	Turn [%]	Statistical Coil [%]
20	35.9	16.5	19.1	29.2
30	36.0	17.8	18.3	28.1
40	35.5	17.1	19.4	28.7
50	34.3	17.3	19.9	29.0
60	23.7	27.6	23.1	25.5
70	7.1	42.2	22.1	28.6
80	6.0	41.2	22.2	30.6

## Data Availability

The files related to the bioinformatics analysis are available at: https://doi.org/10.18150/0D4BNT (Accessed date 10 June 2022).

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
