# Peer review of "Molecular Characterization of a DNA Polymerase from Thermus thermophilus MAT72 Phage vB_Tt72: A Novel Type-A Family Enzyme with Strong Proofreading Activity"

_ijms, 2022, doi:10.3390/ijms23147945_

Round 1

Reviewer 1 Report

DNA polymerases are indispensable enzyme tools in modern molecular biology and have many other life science applications. Dorawa and coworkers are searching for DNA polymerase with unique properties, for developing new genetic tools and biotechnology. In this study, the authors identify a putative type A DNA polymerase gene in thermophilus phage. A predicted structural model is generated, and a serial of functional assays and biochemical/physical experiments are performed using recombinant protein to further characterize this enzyme.

Overall, experiments are well described, and the results support the author’s conclusion. I have only two minor edits/suggestions.  

Minor points

1. I think writing in the following order for the Results section makes more sense (or in the same order as summarized in the abstract): identification of gene, cloning, recombinant expression, then other biochemical/ biophysical/functional characterization. Is there any particular reason for why the authors putting Results section 2.2 with structural prediction and the key catalytic residues mutational effects before the purification of the actual enzyme? Also please try to make the methods subsection in the same order as the results section (for example, at least renumber the section 4.14 as 4.1). Thus, the manuscript would be better organized overall.

2. Figure.2(a), make sure labeled S616 can be seen clearly.

Author Response

Reviewer #1

DNA polymerases are indispensable enzyme tools in modern molecular biology and have many other life science applications. Dorawa and coworkers are searching for DNA polymerase with unique properties, for developing new genetic tools and biotechnology. In this study, the authors identify a putative type A DNA polymerase gene in thermophilus phage. A predicted structural model is generated, and a serial of functional assays and biochemical/physical experiments are performed using recombinant protein to further characterize this enzyme.

Overall, experiments are well described, and the results support the author’s conclusion. I have only two minor edits/suggestions.  

Minor points

  1. I think writing in the following order for the Results section makes more sense (or in the same order as summarized in the abstract): identification of gene, cloning, recombinant expression, then other biochemical/ biophysical/functional characterization. Is there any particular reason for why the authors putting Results section 2.2 with structural prediction and the key catalytic residues mutational effects before the purification of the actual enzyme? Also please try to make the methods subsection in the same order as the results section (for example, at least renumber the section 4.14 as 4.1). Thus, the manuscript would be better organized overall.

ANSWER

We followed for Reviewer’s advice and restructured the Results section accordingly. The same we applied to the Methods section to make the whole text better organized and easy to follow.

  1. Figure.2(a), make sure labeled S616 can be seen clearly.

ANSWER

It has been corrected. Changes in the Result section led to changes in figure numbering. The former Figure 2a is now Figure 7a (revised manuscript).

Reviewer 2 Report

The manuscript from Kaczorowski group focuses on a characterization of a newly expressed DNA polymerase from a phage that infects Thermus thermophilis bacteria. Since DNA polymerases with different properties are widely used as molecular biology tools, characterization of novel thermo-stable proteins is of an immediate interest for various applications. The authors express this new A family polymerase and thoroughly evaluate it by a battery of biochemical and physical-chemical methods. First, based on the structural model of the polymerase, the authors identify the residues that are essential for polymerase and exonuclease active sites and demonstrate their importance by characterization of the mutants. Furthermore, the authors determine temperature, pH and salts concentrations for optimum reaction conditions and show that the protein is a monomer in solution. The authors conclude that the DNA polymerase has a strong proofreading activity and is moderately thermo-stable.

The manuscript is well written, the experiments are technically sound and the data are clearly presented.

I have only a minor comment:

Figure 2b (Table) Only one uncertain digit should be reported for a measurement. For example, 90.6± 1.8 should be re-written as 91 ± 2; 86.3 ± 4.4 should be 86± 4; 96.5 ± 4.1 should be 97± 4.

Author Response

Reviewer #2

The manuscript from Kaczorowski group focuses on a characterization of a newly expressed DNA polymerase from a phage that infects Thermus thermophilis bacteria. Since DNA polymerases with different properties are widely used as molecular biology tools, characterization of novel thermo-stable proteins is of an immediate interest for various applications. The authors express this new A family polymerase and thoroughly evaluate it by a battery of biochemical and physical-chemical methods. First, based on the structural model of the polymerase, the authors identify the residues that are essential for polymerase and exonuclease active sites and demonstrate their importance by characterization of the mutants. Furthermore, the authors determine temperature, pH and salts concentrations for optimum reaction conditions and show that the protein is a monomer in solution. The authors conclude that the DNA polymerase has a strong proofreading activity and is moderately thermo-stable.

The manuscript is well written, the experiments are technically sound and the data are clearly presented.

I have only a minor comment:

Figure 2b (Table) Only one uncertain digit should be reported for a measurement. For example, 90.6± 1.8 should be re-written as 91 ± 2; 86.3 ± 4.4 should be 86± 4; 96.5 ± 4.1 should be 97± 4.

ANSWER

It has been corrected through the text as well in Figure 2b.